# Lipidome Plasticity Enables Unusual Photosynthetic Flexibility in Arctic vs. Temperate Diatoms

**DOI:** 10.3390/md22020067

**Published:** 2024-01-27

**Authors:** Jon Brage Svenning, Terje Vasskog, Karley Campbell, Agnethe Hansen Bæverud, Torbjørn Norberg Myhre, Lars Dalheim, Zoé Lulu Forgereau, Janina Emilia Osanen, Espen Holst Hansen, Hans C. Bernstein

**Affiliations:** 1Norwegian College of Fishery Science, UiT—The Arctic University of Norway, 9037 Tromsø, Norway; lars.dalheim@nofima.no (L.D.); espen.hansen@uit.no (E.H.H.); hans.c.bernstein@uit.no (H.C.B.); 2SINTEF Nord, Storgata 118, 9008 Tromsø, Norway; 3Department of Pharmacy, UiT—The Arctic University of Norway, 9037 Tromsø, Norway; terje.vasskog@uit.no (T.V.); agnethe.h.baverud@uit.no (A.H.B.); torbjorn.n.myhre@uit.no (T.N.M.); 4Department of Arctic and Marine Biology, UiT—The Arctic University of Norway, 9037 Tromsø, Norway; karley.l.campbell@uit.no (K.C.); zoe.l.forgereau@ntnu.no (Z.L.F.); janina.osanen@ntnu.no (J.E.O.); 5The Arctic Centre for Sustainable Energy—ARC, UiT—The Arctic University of Norway, 9037 Tromsø, Norway

**Keywords:** lipidomics, psychrophile, photosynthesis, Arctic, algae, diatom, irradiance

## Abstract

The diatom lipidome actively regulates photosynthesis and displays a high degree of plasticity in response to a light environment, either directly as structural modifications of thylakoid membranes and protein–pigment complexes, or indirectly via photoprotection mechanisms that dissipate excess light energy. This acclimation is crucial to maintaining primary production in marine systems, particularly in polar environments, due to the large temporal variations in both the intensity and wavelength distributions of downwelling solar irradiance. This study investigated the hypothesis that Arctic marine diatoms uniquely modify their lipidome, including their concentration and type of pigments, in response to wavelength-specific light quality in their environment. We postulate that Arctic-adapted diatoms can adapt to regulate their lipidome to maintain growth in response to the extreme variability in photosynthetically active radiation. This was tested by comparing the untargeted lipidomic profiles, pigmentation, specific growth rates and carbon assimilation of the Arctic diatom *Porosira glacialis* vs. the temperate species *Coscinodiscus radiatus* during exponential growth under red, blue and white light. Here, we found that the chromatic wavelength influenced lipidome remodeling and growth in each strain, with *P. glacialis* showing effective utilization of red light coupled with increased inclusion of primary light-harvesting pigments and polar lipid classes. These results indicate a unique photoadaptation strategy that enables Arctic diatoms like *P. glacialis* to capitalize on a wide chromatic growth range and demonstrates the importance of active lipid regulation in the Arctic light environment.

## 1. Introduction

In a marine environment, photoacclimation is crucial to maintaining primary production due to significant variability in the magnitude and wavelength of irradiance within the euphotic zone. These variations stem from the wavelength-selective absorption of water, vertical mixing or stagnation of cells with stratification, extent of cloud cover, as well as scattering and absorption by particulate matter that includes microbial pigmentation [1,2,3,4]. Diatoms are central to marine primary production because of their ability to grow rapidly and export carbon under favorable conditions, like those often present in coastal and upwelling regions, and contribute an estimated 75% of the primary production in these regions [5,6].

The structural lipochemistry of diatoms is characterized by polar glycerolipids, and in a nutrient-replete environment, thylakoid membranes incorporate four major lipid classes: monogalactosyldiacylglycerol (MGDG), digalactosyldiacylglycerol (DGDG), sulfoquinovosyldiacylglycerol (SQDG) and phosphatidylglycerol (PG) [7]. During periods of phosphate limitation, phospholipids may be substituted by non-phosphorous betaine lipids [8].

As structural components of the thylakoid membranes, these lipids maintain the photosynthetic capacity of chloroplasts, which rely on pigments to capture energy from light. The main light-harvesting pigments in diatoms are chlorophyll *a* (Chl *a*), chlorophyll *c* (Chl *c*) and the carotenoid fucoxanthin, which together form the light-harvesting complexes in diatoms known as the fucoxanthin chlorophyll protein complexes (FCPs) embedded in the thylakoid membrane. FCPs replace the light-harvesting complexes (LHCs) of plants and photosynthetic bacteria and have adapted to the blue-green radiation characteristic of aquatic environments by incorporating higher ratios of fucoxanthin to chlorophyll compared to LHCs [9]. In addition to fucoxanthin, other carotenoids such as β-carotene contribute directly to the light harvesting of photosynthetically active radiation (PAR), while xanthophylls (oxygenated carotenes) such as diatoxanthin and diadinoxanthin contribute towards photoprotection through different mechanisms such as the de-excitation of single excited states of chlorophyll, prevention of singlet oxygen release by the quenching of chlorophyll’s triplet state, and by acting as antioxidants [10]. Diatoms acclimate to the intensity and wavelength distributions of irradiance by regulating the intracellular concentrations of these functional lipids and lipophilic pigments directly as structural modifications of protein–pigment complexes, in so-called chromatic acclimation, and through the active cycling of xanthophylls in photoprotection [11,12,13,14]. Structural lipids also contribute to diatoms’ ability to cope with changing irradiance: As diatoms alter the relative abundance of pigment complexes in their thylakoid membrane in response to changing irradiance, the fluidity of the membrane is maintained by altering the ratio of the non-bilayer-forming lipid MGDG and the bilayer-forming lipid DGDG [15]. Thylakoid membrane lipids also contribute to the photoprotective mechanisms of cells by acting as solvents for xanthophyll cycle pigments and by shifting carbon allocation towards storage lipids to create energy sinks which safely dissipate excess light energy during photosynthesis [16,17].

The potential for variability in the light environment of marine diatoms, and thus the need for photoacclimative responses, is greater in polar regions due to the strong seasonality (i.e., polar day and night cycles), high frequency of cloud cover, presence of attenuating sea ice cover, as well as high solar zenith angle of downwelling [18,19]. As a result, numerous studies have demonstrated enhanced dark survival and low irradiance requirements of Arctic diatoms [20,21], in addition to niche-dependent nonphotochemical quenching strategies [22]. However, chromatic adaptations in the Arctic marine lipidome remain largely unexplored, and the balances and interactions between lipophilic components, including pigments, in response to spectral irradiance as a function of photon wavelength remain poorly understood. We expect that the lipidome in combination with photosynthetic pigments plays a central role in the ability of Arctic marine diatoms to deal with extreme variations in ambient irradiance.

The aim of this study was to test the hypothesis that diatoms adapted to Arctic marine environments will display distinct acclimations that regulate their functional lipids and pigment composition to accommodate a wider photic niche and thereby maintain photosynthetic growth in a more complex and changing light environment, as compared to those adapted to lower-latitude (mesophilic) light conditions. We tested this hypothesis by comparing the lipidome, including pigments, and photophysiology between the Arctic diatom *Porosira glacialis* and mesophilic control species *Coscinodiscus radiatus*. These two centric diatoms were each cultivated under red (621 nm), blue (457 nm) and white light, at temperatures typical of the Arctic (8 °C) and temperate (20 °C) oceans, respectively. Growth was measured as in vitro chlorophyll *a* (Chl *a*), and the effect of irradiance quality on photosynthesis was measured using photosynthesis–irradiance curves (P–I curves). Lipids and pigments were identified and quantified based on raw data acquired by LC-MS, using the software LipidSearch ver. 4.2 and Compound Discoverer ver. 3.3 by Thermo Fisher Scientific, Waltham, MA, USA. To our knowledge, this represents the first study that couples untargeted lipidomics with pigment analysis and comparative photophysiology between different diatom species. Understanding how the lipidome mediates photosynthetic flexibility in *P. glacialis* and *C. radiatus* adds new knowledge on the fundamental biology of Arctic vs. temperate diatoms and sheds light on Arctic diatoms’ success in a more complex light environment.

## 2. Results

### 2.1. Differential Responses in Fatty Acid Esterification Highlight Species-Specific Responses to Light

Light quality revealed distinct responses in the lipid composition of the two diatom species. In both diatoms, blue light resulted in an increased inclusion of long-chain polyunsaturated fatty acids (fatty acids with ≥20 carbons and ≥3 double bonds, LC-PUFAs) in the lipidome; however, in *P. glacialis*, these were preferentially esterified to neutral TGs, while *C. radiatus* targeted structural phospho- and galactolipids for LC-PUFA esterification. Eleven lipid classes were characterized in the two diatom species, containing a total of 44 and 45 unique fatty acids in *P. glacialis* and *C. radiatus*, respectively. In total, more than 300 unique lipid molecules from each sample passed the inclusion thresholds set by the LC-MS method.

Light quality affected the composition of the main lipid classes, fatty acids and molecular species in the two diatoms; however, the lipid molecules targeted for fatty acid esterification differed fundamentally between the two species. In the Arctic diatom, *P. glacialis*, light quality did not have a significant impact on LC-PUFA esterification in the polar structural lipids, and instead, triglycerides (TGs) were the main targets of fatty acid esterification. In comparison to red light, blue and white light shifted the composition of fatty acyl groups esterified to the neutral TGs towards the LC-PUFAs, primarily C20:5 and C22:6 (Figure 1A,C). This effect was strongest under blue light, with increased LC-PUFA contribution in the TG in comparison to white light (Figure 1B). A total of 22, 4 and 37 differentially abundant lipid compounds exceeded the significance thresholds of *p* < 0.5 and fold change ≥ 2 in *P. glacialis* for red vs. white, blue vs. white and blue vs. red light, respectively (Table 1).

Red and blue light also affected the molecular species of the structural lipids MGDG, PE, PG and PC in *P. glacialis*, but without any apparent common trend. As a proxy for relative abundance, the total peak areas of lipid classes were also affected by light quality in *P. glacialis*. In this case, red light resulted in increased inclusions of the polar lipid classes MGDG, PE and PG in comparison to blue and white light (Figure 2). The mean peak area of TG was unaffected by light quality, but the contributions of both DG and MG were higher under red light. Blue light also resulted in statistically significantly higher inclusions of the LC-PUFAs C20:5 and C22:6.

In the mesophilic diatom *C. radiatus*, light quality influenced the molecular composition of the polar phospho- and galactolipids. In comparison to white light, red light resulted in a lower contribution of the C16 and LC-PUFAs and an increased contribution of C18 fatty acids in the polar lipid classes (Figure 3A), while blue light drove the inclusion of C16, C18 and the LC-PUFAs C20:5 and C22:6 in the polar lipid classes (Figure 3B). The greatest difference was observed between samples cultivated under blue and red light. In this case, blue light resulted in a large increase in the contribution of LC-PUFA acyl groups in MGDG, PG, PC, PE and their corresponding lysophospholipids (Figure 3C). A total of 17, 20 and 59 differentially abundant lipid compounds exceeded the significance thresholds of *p* < 0.5 and fold change ≥ 2 in *C. radiatus* for red vs. white, blue vs. white and blue vs. red, respectively (Table 1). The relative abundances of the individual lipid classes and fatty acids also showed an increased contribution of PE, PC, lysophosphatidylglycerol (LPG), lysophosphatidylethanolamine (LPE) and lysophosphatidylcholine (LPC), as well as an increase in the LC-PUFAs C20:5, C22:4 and C22:6 in the samples cultivated with blue light (Figure 4). The contributions of the thylakoid-associated lipids MGDG and PG were highest in blue and white light, respectively, but did not significantly differ between these light qualities in *C. radiatus*.

The results show that the chromatic environment exerts a unique influence on lipid composition and fatty acid allocation in the Arctic diatom *P. glacialis* and the mesophile *C. radiatus*, highlighting the importance of considering environmental factors in understanding diatom metabolism.

### 2.2. Pigment Analysis Demonstrates Species-Specific Responses in Pigment Composition as a Function of Light Quality

Light quality influenced the relative pigment content in the two diatoms; red light increased Chl *a*, Chl *c1* and fucoxanthin in the Arctic *P. glacialis*, and blue light increased Chl *a*, Chl *c1*, Chl *c2*, beta-carotene and diatoxanthin in the mesophilic *C. radiatus* (Figure 5 and Figure 6, respectively).

The AcquireX acquisition workflow identified nine pigments with a high level of confidence from the in-house mzVault database. Beta-carotene, Chl *a*, chlorophyll *c1* (Chl *c1*), chlorophyll *c2* (Chl *c2*), diadinoxanthin, diatoxanthin, fucoxanthin and pheophytin a were identified in both diatoms, while violaxanthin was only found in *C. radiatus*. Light quality influenced the relative content of pigments in the two diatoms. In *P. glacialis*, red light resulted in an increased inclusion of the main light-harvesting pigments, Chl *a*, Chl *c1*, and fucoxanthin, compared to both blue and white light (Figure 5).

Light quality did not have a significant effect on the relative content of Chl *c2*, beta-carotene, diadinoxanthin and diatoxanthin in *P. glacialis*. In *C. radiatus*, blue light resulted in a higher relative content of Chl *a*, Chl *c1*, Chl *c2* and beta-carotene and a higher inclusion of diatoxanthin in comparison to white light (Figure 6). Blue light did not have a significant influence on the relative content of diadinoxanthin, fucoxanthin and violaxanthin in comparison to white light, and red light did not significantly influence any of the identified pigments in *C. radiatus* in comparison to white light. The ratios of Chl *a* to pheophytin a based on the areas under the curve of the chromatograms were 0.74, 0.73 and 0.81 for *P. glacialis* and 0.52, 0.90 and 0.43 for *C. radiatus* under red, blue and white light, respectively. The results indicate species-specific strategies in chromatic acclimation in diatom’s adaptation to different light conditions.

### 2.3. The Arctic P. glacialis Utilizes Red Light Effectively to Power Carbon Fixation

The Arctic diatom *P. glacialis* displayed similar growth and consistent photosynthetic rates across light conditions, while the mesophilic diatom *C. radiatus* achieved greater maximum photosynthetic rates in blue and white light in comparison to red light. The growth physiology of both strains in response to their chromatic environment was measured as changes in Chl *a* during log phase growth and gross primary production measurements. The Arctic strain *P. glacialis* achieved the highest specific growth rate in red light (μ = 0.017 h^−1^, *p* < 0.05) and transitioned towards stationary growth after 160 h, as observed in the leveling of the curve plots displaying the in vitro Chl *a* content (Figure 7).

The in vitro Chl *a* increased in concentration corresponding to average doubling times of 45.5, 50.1 and 40.2 h in *P. glacialis* in white, blue and red light, respectively. The mesophile *C. radiatus* achieved the highest specific growth rate (μ = 0.013 h^−1^) and in vitro Chl *a* content in white light, followed by red and blue light (*p* < 0.05, Figure 7). The increase in in vitro Chl *a* over time corresponded to doubling times of 54.0, 64.0 and 64.8 h in *C. radiatus* in white, blue and red light, respectively. The amount of Chl *a*/cell was stable across all treatments in both diatoms at the start of the experiment, following the 7-day acclimatization period (Table 2).

The chromatic quality of light also affected gross primary production and photophysiology in the two diatom strains (Figure 8), as determined from P–I curves measuring ^14^C-labelled CO_2_ uptake rates under the function of red, blue and white light treatments. In *P. glacialis*, the maximum photosynthetic rate (or maximum rate of carbon fixation during photosynthesis) in the absence of photoinhibition (P_s_^B^) was consistent across all treatments, while the mesophilic diatom *C. radiatus* achieved greater maximum photosynthetic rates in blue and white light in comparison to red light. The photosynthetic efficiency (α^B^), which is the amount of light energy converted into chemical energy, was also lower in *C. radiatus* under red light. Beyond this exception, the photosynthetic efficiencies were similar across all treatments in both species. The value for photoacclimation (I_k_), which is a measurement of the ability to adjust to changes in irradiance, was highest in red light in both species.

## 3. Discussion

### 3.1. Blue Light Drives Increased Inclusion of LC-PUFAs

The composition of lipid molecular species indicates that blue light drives the increased inclusion of LC-PUFAs in the lipidome of both *C. radiatus* and *P. glacialis*, shifting the differential abundance of molecular species towards longer, more unsaturated fatty acids. Similarly, the blue photons are most likely responsible for this same effect in both species observed under white light, as white LEDs contain both blue and red photon wavelengths. However, while the Arctic strain *P. glacialis* targeted neutral TGs for LC-PUFA esterification, the mesophilic strain *C. radiatus* preferentially bound these fatty acids to the polar, structural lipids MGDG, PG, PC and PE. A recent study on *Nannochloropsis oceanica* found that blue light induced fatty acid desaturases (FADs) involved in LC-PUFA biosynthesis and maintained LC-PUFA levels in cells, while red light resulted in a reduction in eicosapentaenoic acid (EPA) content [23]. This mechanism was presumably mediated by light-sensing proteins specific to stramenopiles called aureochromes, which can sense blue light and, in this case, were shown to act as transcription factors for genes involved in LC-PUFA biosynthesis. Our results are consistent with this finding; however, the mechanisms that control the expression of fatty acids and allocation to neutral lipids in *P. glacialis* and polar lipids in *C. radiatus* are unclear and, to our knowledge, have not been previously demonstrated in diatoms. Triglyceride accumulation following growth limitation in *P. glacialis* is an unlikely explanation for this observation as the total amount of TGs was independent of light quality in the polar strain. However, it is possible that TG in this case functions as a reservoir for LC-PUFAs in *P. glacialis* for future use when these fatty acids are superfluous in other functional lipids such as thylakoid membrane lipids. A recent study on the model diatom *Phaeodactylum tricornutum* found that blue cultivation light resulted in a higher production of saturated fatty acids bound to TAGs, while red cultivation light increased the cell’s content of the LC-PUFAs hexadecatrienoic acid (HTA) and EPA [24]. Although inconsistent with our findings, these results also demonstrate the diatoms’ underlying ability to regulate fatty acid allocation in response to the wavelength of light.

### 3.2. Structural Lipids and Pigments Unlock Photic Niches of Diatoms

The two diatom lipidomes also displayed a dependence on light quality during log phase growth, in parallel with changes to pigmentation. The greatest relative content of PGs, the only major phospholipid in thylakoid membranes among oxygenic phototrophs, coincided with the light qualities that also generated the highest specific growth rates and P^B^_m_ in each diatom strain, i.e., white light in *C. radiatus* and red light in *P. glacialis*. While the glycolipids of the thylakoid membranes are primarily distributed in the lipid bilayer, PG is specifically allocated to the embedded photosystems in higher plants and cyanobacteria, where they are involved in electron transport processes [11,25]. Our results also indicate a similar function of PG in the diatom lipidome, as the relative content of PG was positively correlated with the photophysiological health (i.e., highest specific growth rates and P^B^_m_ of both *C. radiatus* and *P. glacialis*). The contribution of MGDG, the only other major thylakoid lipid that was detected in this study, was highest in blue light in *C. radiatus* and red light in *P. glacialis* (Figure 2 and Figure 4), which coincides with the light qualities that also contained the highest relative amounts of the main light-harvesting pigments Chl *a*, Chl *c1,* Chl *c2* and fucoxanthin in both diatom strains (Figure 5 and Figure 6). This is likely a result of the important functions that glycolipids, particularly the non-bilayer-forming lipid MGDG, serve in stabilizing pigment–protein complexes and photosystems in thylakoid membranes [26,27].

In higher plants, MGDG and PG have also been linked to Chl *a* through biosynthesis pathways, where they regulate the activity of protochlorophyllide oxidoreductase [28]. The observed similarity between the relative amounts of thylakoid lipids, photosynthetic pigments and photophysiological conditions in this study are therefore unsurprising. However, if these molecules facilitate photosynthetic growth as a function of the chromatic quality of light, our findings indicate differences in their ability to occupy different photic niches in marine ecosystems by adapting the composition of both structural lipids and pigments to specific wavelengths of PAR. In this case, the results show that the Arctic strain *P. glacialis* can acclimate its composition of thylakoid-associated lipids in coordination with pigment allocation to utilize red light more efficiently than the mesophilic *C. radiatus*, thereby gaining an ecological advantage that allows high growth rates across a potentially wider range of conditions.

### 3.3. Red Light Reveals Photosynthetic Flexibility in P. glacialis

The Arctic strain *P. glacialis* achieved the highest growth rates under red light and a stable maximum photosynthetic rate across all light qualities. This is despite a photophysiology that is generally considered (in diatoms) to be more adapted to the blue light absorption characteristic of a marine environment and the understood ability of blue light to drive higher growth rates and photosynthetic efficiencies compared to red light during exponential phases of growth [29,30,31]. In the mesophilic strain *C. radiatus*, red light resulted in reduced specific growth rate and maximum photosynthetic rate—as determined by in vitro Chl *a*—compared to blue and white light.

In contrast to *C. radiatus*, *P. glacialis* displayed stable ratios of Chl *a* to pheophytin *a*, independent of light qualities. This ratio is frequently used as a measurement of the physiological condition of phytoplankton. However, despite the lower maximum photosynthetic rate (PsB) and photosynthetic efficiency (αB) of *C. radiatus* cultivated under red light, there were no significant differences in the Chl *a*-to-pheophytin ratio, or in the relative content of the main photosynthetic pigments and structural lipids associated with the thylakoid membranes when comparing red to white light. The only discernable effect of red light on *C. radiatus* that may have affected its photophysiology was the reduced LC-PUFA allocation to structural lipids in comparison to white (and blue) light, which was not observed in *P. glacialis*. In addition to regulating the phase transition of membranes and scavenging reactive oxygen in lipid peroxidation, it is generally assumed that LC-PUFAs contribute to the photosynthetic function of algae due to their frequent presence in the thylakoid-associated lipids MGDG, DGDG and SQDG [32]. If the lowered maximum photosynthetic rate and photosynthetic efficiency of *C. radiatus* under red light can be attributed to the composition of fatty acids, our results indicate that the *C. radiatus* photophysiology may be dependent on blue photons to maintain sufficient biosynthesis of LC-PUFA to function effectively.

These results suggest an unusual photosynthetic flexibility in *P. glacialis* with ecological relevance in a dynamic light environment. *P. glacialis* also displayed significantly higher growth rates and a total increase in Chl *a* over time compared to *C. radiatus*, independent of light quality, despite a comparatively much lower cultivation temperature. *C. radiatus* is considerably larger than *P. glacialis*, with diameters of approximately 90 and 40 μm, respectively. While growth rates are often positively correlated to temperature within a given species’ natural temperature range, as previously demonstrated in *P. glacialis* [33], microalgal growth rates are species-specific and larger diatoms typically display lower rates of cell division [34,35]. However, the capacity for rapid growth at comparatively low temperatures and wide chromatic ranges demonstrates the important role of *P. glacialis* as a primary producer in highly productive areas. This observation also provides insight into the potentially large impact that climate change and subsequent shifts in species composition may have on primary production in warming polar environments with reduced ice cover.

## 4. Materials and Methods

### 4.1. Algal Strains and Growth Conditions

*Coscinodiscus radiatus* (CCAP 1013/11) was acquired from the Scottish Association for Marine Science (SAMS). The strain identified as *P. glacialis* UiT201 was isolated using conventional single-cell isolation with a drawn Pasteur pipette from water samples collected in the Barents Sea (N 76°27.54′, E 33°03.54′) and identified using SEM imaging and rcbL and 16S gene barcoding [36,37]. Cultivation was performed in four-liter polycarbonate bottles (Nalgene, Rochester, NY, USA) incubated at temperatures emulating psychrophilic (8 °C, *P. glacialis*) and mesophilic (20 °C, *C. radiatus*) conditions. Cultures were grown in Guillard’s F/2 marine water enrichment with added silicate (Sigma-Aldrich, St. Louis, MO, USA). The choice of growth temperature was based on the diatoms’ natural temperature range in a typical Arctic and mesophilic light environment. Illumination was supplied by LED light strips (Co/tech Model 36-7237; Surrey, UK) set to red, white or blue light, and they were adjusted to 35 μmol m^−2^ s^−1^ using an LI-250A light meter (LI-COR, Cambridge, UK) coupled with a Walz US-SQS/L sensor (Heinz Walz GmbH, Effeltrich, Germany). The peak wavelengths for each light treatment were 629 nm for red light, 457 nm for blue light, and 457, 521 and 629 nm for white light, recorded using a BLACK-Comet Fluorescence Spectrophotometer (StellarNet, Tampa, FL, USA).

### 4.2. Experimentation

Prior to the initiation of the experiment, inocula of *P. glacialis* and *C. radiatus* were acclimatized to their respective red, blue or white light environments for 7 days, before dilution, at a starting concentration of 1 × 10^6^ cells/L and 5 × 10^5^ cells/L for *P. glacialis* and *C. radiatus*, respectively. We chose to use different initial cell concentrations for two main reasons: first, the starting concentrations were set high enough to prevent the diatoms from entering the lag phase upon initiation of the experiment, based on previous experience with these strains, and second, the differing cell size to achieve approximately the same amount of biovolume. Specific growth rates were calculated based on changes in in vitro chlorophyll *a* (Chl *a*) using the method developed by Holm-Hanssen and Riemann [38], modified to use ethanol as the solvent, measured as fluorescence on a Trilogy fluorometer (Turner, Sunnyvale, CA, USA). Chl *a* measurements were used to calculate specific growth rates (μ) by fitting ln (X)=µt via linear regression on the semi-log-transformed time-series data, where *X* represents the time-dependent Chl *a* concentration. The cultures were harvested after seven days of growth.

### 4.3. Harvesting for Pigment and Lipid Extraction

The cultures were harvested by gravity filtration through a 10 μm phytoplankton net (KC Denmark AS, Silkeborg, Denmark) and subsequent centrifugation at 2000× *g* for 4 min (Heraeus Multifuge 1S-R; Danau, Germany) in 50 mL centrifugation tubes (Corning Science, Reynosa, Mexico). The supernatant was discarded before the samples were flash-frozen in liquid nitrogen and stored at −80 °C. The raw data including all lipid species and pigments that were included in the analysis are available in the open science framework associated with this study (see Data Availability Statement).

### 4.4. Lipid and Pigment Extraction

Lipids and pigments were extracted from lyophilized material as described in [39]. Briefly, 100 mg of material was added in 20 volumes (2 mL) of DCM/MeOH (2:1 *v*/*v*) and 5% NaCl, respectively, and centrifuged at 2000× *g* for 5 min before transferring the organic phase to pre-weighed 4 mL glass vials and evaporating the solvents under nitrogen. The sample lipid content was gravimetrically determined. The lipids were resuspended in 2-propanol to a concentration of 0.5 mg/mL and transferred to Waters 12 × 32 mm screw top vials with cap and pre-slit PTFE-silicone septa (Waters, Milford, MA, USA).

### 4.5. LC-HRMS Analysis

A description of the settings used in the LC-MS analysis can be found in the Appendix A. The LC-MS analysis of lipids and pigments was performed on a Vanquish Horizon UHPLC coupled to an Orbitrap ID-X (Thermo Fisher Scientific, Waltham, MA, USA). For chromatographic separation, a Waters Acquity UPLC (Waters, Milford, MA, USA) BEH C18 Column (2.1 mm × 100 mm, 1.7 µm) was employed, operating at 60 °C. The binary solvent system included a constant flow rate of 0.6 mL min-1 with a mobile phase consisting of solvents A (50% water/50% acetonitrile) and B (49.5% 2-propanol/49.5% acetonitrile/1% water). Both solvents were supplemented with 1 mM ammonium formate (NH4FA) and 0.01% formic acid (FA). A gradient program was optimized as the following: 0–2.5 min, 30% B; 2.5–20 min, 30–75% B; 20–25 min, 75–95% B; 25–30 min, 95% B (total run time of 32.5 min). The autosampler tray temperature was 5 ℃ and the utilized injection volumes were 2 µL and 5 µL for full scans and MS^n^ scans, respectively. The mass spectrometer used a heated electrospray ionization (H-ESI) interface operated in positive ionization mode with the following parameter settings: electrospray voltage, 3.5 kV; ion transfer tube temperature and vaporizer temperature, 350 °C; sheath gas, 60 arbitrary units (a.u.); auxiliary gas, 15 a.u; and sweep gas, 2 a.u. Full scans were performed by the orbitrap detector with a resolution at 120 000 and a 250–1500 *m*/*z* scan range (Appendix A). An intensity threshold filter of 1.0 × 10^5^ allowed data-dependent higher-energy C-trap dissociation (HCD) MS^2^ fragmentation operating at a stepped collision energy mode (25, 30, 35%) (Appendix A). Identification of PC was performed using collision-induced dissociation (CID) MS^2^ analysis by inclusion of a targeted HCD product ion trigger corresponding to the phosphatidyl head group (*m*/*z* 184.0733 ± 10 ppm) (Appendix A). Triglycerides were identified using CID MS^3^ analysis by inclusion of the ammonium adduct of fatty acids ± 10 ppm (Appendix A).

### 4.6. Acquisition Workflow for Untargeted Analysis

Untargeted lipidomics analysis was performed with the AcquireX Data Acquisition Technology (Thermo Fisher Scientific, Waltham, MA, USA), using a deep scan (DS) data-dependent acquisition (DDA) method. The acquisition mode generated an exclusion and inclusion list based on full-scan data of the 2-propanol system blank and a pooled average of all algae samples to be analyzed, respectively. The pooled sample was then injected to produce exhaustive MS^n^ data by automatically updating the exclusion and inclusion lists with an iterative cycle of 5. The acquired MS^n^ data were used for the identification of lipids and pigments, while subsequent injections of all sample extracts in the full scan ensured data quality for relative quantification. The injection sequence and workflow parameters are depicted in Appendix A. During the method development, a mixed lipid standard (LightSPLASH LIPIDOMIX; Avanti Polar Lipids, Alabaster, AL, USA) containing 13 lipids (15:0-18:1 PC, 18:1 LPC, 15:0-18:1 PE, 18:1 LPE, 15:0-18:1 PG, 15:0-18:1 PI, 15:0-18:1 PS, 15:0-18:1-15:0 TG, 15:0-18:1 DG, 18:1 MG, 18:1 cholesterol ester, d18:1-18:1 sphingomyelin, C15 ceramide (d18:1%15:0)) and a mix of MGDG, mainly 16:3-18:3 and 18:3-18:3 (Avanti Polar Lipids), were used to confirm identification in LipidSearch.

### 4.7. Spectral Library Creation for Pigment Identification

A spectral library containing 17 pigments was built by utilizing a targeted ddMS^2^ (data-dependent MS^2^ acquisition) method to analyze a pooled sample of the pigment standards (see Appendix A for masters can configuration, targeted mass list and HCD MS^2^ settings, respectively). The mass-to-charge ratio of the included pigments including their radicals and proton adducts are displayed in Appendix A. Mass spectra were imported with both precursor ions and characteristic fragments of the identified pigments using mzVault v2.3 (Thermo Fisher). The most abundant adduct for each pigment standard was considered and the most intense fragmentation spectrum was automatically selected by the software’s algorithm. All spectra included in the library were manually reviewed with regards to retention time in alignment with preliminary runs, the presence of a molecular ion, a sufficient number of fragments and signal intensity.

### 4.8. Lipid Identification by LipidSearch

This study used LipidSearch to identify lipid molecular species. The software functions by importing raw LC-MS data and identifying matches in an online theoretical database (in silico) based on the mass-to-charge ratio and the presence of fragment ions in the MS2 and MS3 spectra of these compounds. The software also performs an evaluation of each match based on these parameters and provides a score from A to D, which is used to express the confidence or quality of a match between observed data and database entries. While a score of A or B usually implies strong confidence, a C or D score indicates that a given compound deviates from one or more theoretical parameters, or in cases where fragments are missing in the sample data. This does not necessarily imply a failed identification. Missing fragment data (due to, e.g., low concentration of the sample compound) can yield a low score despite a correct identification by the software. A manual inspection of the results is therefore beneficial. In this study, each match was inspected and either excluded or included in the results based on the quality of the peak shape and its effect on integration, the relative standard deviation of the integrated peaks (<30% was used as the threshold for inclusion), correlation between the observed retention time in the pooled ID sample and the individual samples (0.1 min threshold), and the isolation window purity at the MSn event (±0.75 Da.) Identifications with C–D scores in LipidSearch were only included if the isolation window contained no interference peaks, while a <10% threshold was used for A–B scores.

### 4.9. Measurement of Photophysiology

Photosynthesis–irradiance (P–I) curves of gross primary production were determined using a previously described ^14^C (4 µCi ml^−1^) tracer [40] and the incubation method detailed by Ref. [41]. Here, a single subsample from each light quality and species (*n* = 6) was aliquoted into ten 60 mL polystyrene culture flasks (Corning) and incubated for 3 h in an incubation chamber, across a range of light intensities from 6 to 225 µmol m^−2^ s^−1^, depending on the distance from the LED light source (Co/tech Model 36-7237). The maximum light intensity was limited by the LED light source, which could not provide more than 225 µmol m^−2^ s^−1^ of red light. In this set up, the flask closest to the light source received the highest light intensity, and the flask at the back of the chamber received the lowest light intensity, resulting in a single series of 10 measurements per treatment. We were only able to make single series of measurements for each species and light quality, as we only had two incubation chambers and a single series of measurements took approximately 4 h including preparation, which limited our ability to perform replicates within a reasonable time frame to compare data. Two flasks were also incubated in darkness with 3-3,4-dichlorophenyl-1,1-dimethylurea (DCMU) during a given P–I incubation to correct for the osmotic uptake of ^14^C [42]. The wavelength of the incubation light source (i.e., blue, red, white) and temperature used in P–I incubations were modified to match the wavelength and temperature used in experimental growth (see Section 2.1). It follows that three light treatments of *P. glacialis* at 8 °C and *C. radiatus* at 20 °C were incubated, for a total of six treatments. The average light intensity (*n* = 3) was measured in the center of each bottle position using a PAR probe (Walz US-SQS/L). Following incubation, samples were filtered onto 25 mm GF/F filters (Whatman, Maidstone, UK) before acidification with 0.5 N HCl, drying for at least 24 h and finally adding a 10 mL EcoLume scintillation cocktail. The radioactivity of the filters was counted (Tri-Carb 2900 TR) after 72–96 h of extraction. Photophysiological parameters were determined by fitting the P–I saturation curves in the absence of photoinhibition [43], as given by Equation (1):(1)$$P^{B}=P_{s}^{B}\left(1-e^{-\frac{\alpha^{B}I}{P_{s}^{B}}}\right)$$
where $P^{B}$ is the photosynthetic rate at irradiance *I*, $P_{s}^{B}$ is the maximum photosynthetic rate in the absence of photoinhibition and $\alpha^{B}$ is the initial slope of the P–I curve. The following parameters were calculated using this model and standardizing to Chl *a* in-solution: photosynthetic efficiency $\alpha^{B}$ (mg C [mg Chl *a*]^−1^ l^−1^h^−1^ (µmol photons m**^−^**^2^s^−1^)^−1^), maximum photosynthetic rate$\;P_{s}^{B}$ (mg C [mg Chl *a*]^−1^ l^−1^h^−1^) and photoacclimation index I_k_ (µmol photons m**^−^**^2^s^−1^).

### 4.10. Data Processing

Identification data (MS^n^) were aligned with the full-scan data of all samples by the software LipidSearch 4.2 and Compound Discoverer 3.3, supplemented with the mzVault pigment database (Thermo Fisher Scientific, Waltham, MA, USA) for identification and relative quantification of lipids and pigments, respectively. For a detailed description of how LipidSearch functions, see [44]. See Appendix A for configurations of identification, quantitation, filters, adduct selection and alignment settings, respectively. Additionally, all peaks were manually integrated to ensure uniformity of inclusion, and the relative abundances between treatments was inferred from the normalized total peak areas. The Compound Discoverer workflow and a description of each node are provided in Appendix A. Briefly, the post-acquisition data were processed through a workflow which performs compound detection, compound grouping, prediction of elemental composition, removal of chemical background and automatic compound identification by matching with the in-house mzVault pigment library (ddMS^2^). Parameters considered for positive identification of pigments were a >70% mzVault match score for MS^2^ spectra and an accurate mass (<5 ppm) and retention time. The ratio of Chl *a* to pheophytin *a* was calculated based on the area under the curve of the chromatograms.

### 4.11. Data Presentation and Statistics

All analyses and graphs were prepared using R v3.6.1 “Action of the Toes” [45]. P–I curves were plotted using the “phytotools” package [46]. Means of specific growth rates, Chl *a,* and peak areas of lipid classes were compared with the pairwise Tukey test, assuming a normal distribution. Means were determined to be homogenous at a significance level of >0.05. Lipid class compositions were derived from the total peak areas of molecular species corresponding to unique lipids, e.g., all molecular species of MGDG. Fatty acid compositions were derived from all instances of unique fatty acids and quantified as the total peak areas of their associated lipid classes. Differential abundance analyses of lipid molecular species were performed and visualized using the “EnhancedVolcano” package [47]. Here, log2 fold change was calculated as the log2 ratio of peak areas of molecular species, and *p*-values were determined using pairwise t-tests comparing peak areas of molecular species and –log10 transformed such that a larger value indicated a more significant difference. The relative pigment composition was calculated from the total peak area in the chromatograms and presented as the mean total area in box plots. All raw data, including the methods for hypothesis testing, descriptive statistical procedures and plotting, are included in the R markdown supplied with this study (see Data Availability Statement).

## 5. Conclusions

This study investigated the ability of two centric diatoms, the Arctic strain *P. glacialis* and the mesophilic strain *C. radiatus*, to regulate the functional diversity of their lipidome and pigment composition to maintain production in a changing light environment. This was performed by analyzing Chl *a* specific growth rates, carbon assimilation, pigment composition and lipidomics during log phase growth in red, blue and white light. While both diatoms were able to maintain growth across all treatments, the chromatic quality of light influenced carbon assimilation, pigment composition and the lipidome differentially in the two species. The highest specific growth rate, P^B^_m_ and relative abundance of their thylakoid-associated lipids coincided with red light in *P. glacialis* and white light in *C. radiatus*. *P. glacialis* maintained a more consistent photophysiology—as assayed by photophysiology parameters—across all treatments and generally achieved higher growth rates as compared to *C. radiatus*. In both diatoms, blue light resulted in an increased inclusion of LC-PUFAs in the lipidome; however, in *C. radiatus*, these were preferentially esterified to the structural phospho- and galactolipids, while *P. glacialis* targeted the neutral TGs for LC-PUFA esterification. Our results indicate that the Arctic diatom *P. glacialis* can effectively adapt its lipidome in coordination with photosynthetic pigments towards spectral differences in the available PAR to allow for a wide chromatic growth range, thereby providing a likely ecological advantage within highly variable chromatic environments. In comparison, the mesophile *C. radiatus* was dependent on blue or white light to maintain a photosynthetic efficiency and maximum photosynthetic rate, possibly due to reduced LC-PUFA allocation in polar lipid classes in the absence of blue light, which was not observed in *P. glacialis*. The independence of blue light in maintaining photosynthetic activity indicates a unique adaptation in Arctic diatoms’ ability to cope with relatively extreme environments with respect to light availability.

## Figures and Tables

**Figure 1 marinedrugs-22-00067-f001:**
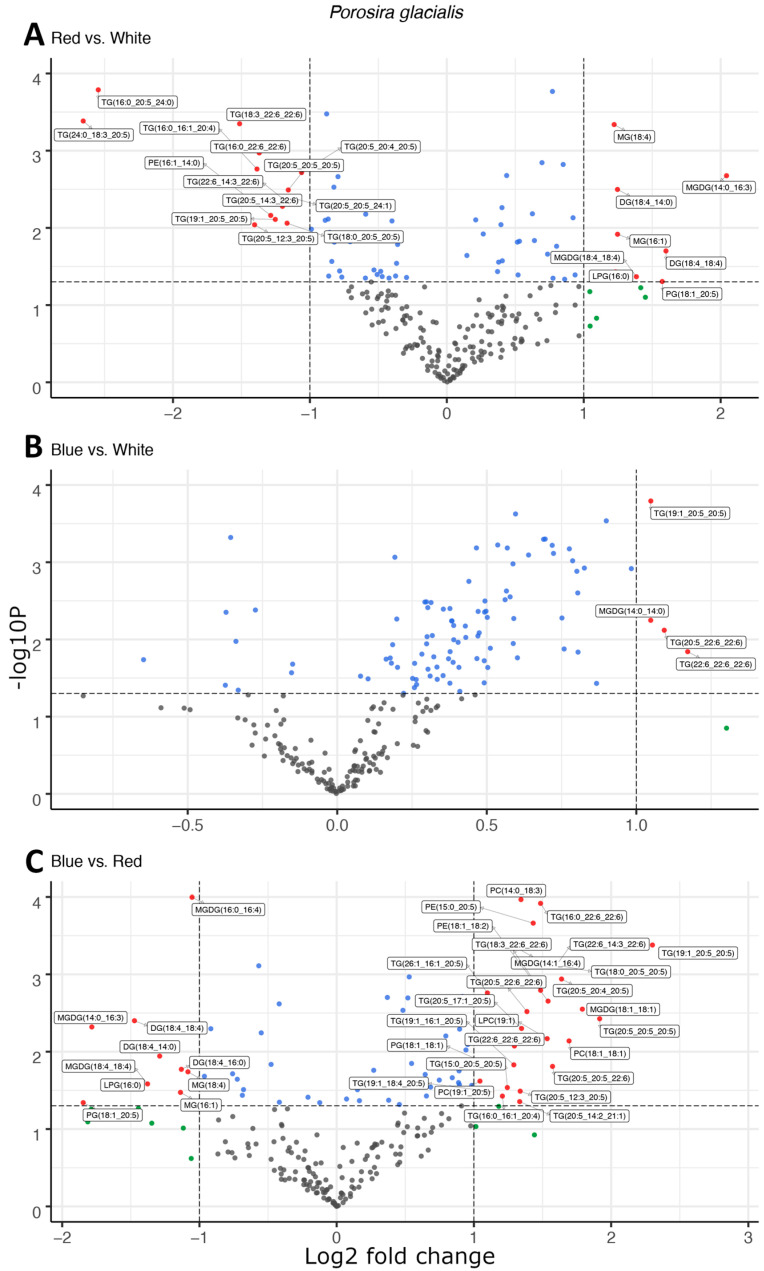
Light quality drives the esterification of long-chain polyunsaturated fatty acids in neutral triglycerides observed as the differential abundance of molecular species in volcano plots of red and white light (**A**), blue and white light (**B**), and blue and red light (**C**) in *Porosira glacialis*. The *y*-axis is defined as the negative log10 of the *p*-value, such that a higher value represents a lower *p*-value. The horizontal and vertical lines represent thresholds to determine significant values (*p* < 0.05 and fold change ≥ 2). The green dots represent data points with a significant fold change, blue dots represent data points with a significant *p*-value and red data points exceed both significance thresholds (*p* < 0.05, fold change ≥ 2). Data points represent the average peak area for each lipid molecule, *n* = 3. Diglyceride (DG), lysophosphatidylcholine (LPC), lysophosphatidylglycerol (LPG), monogalactosyldiacylglycerol (MGDG), monoglyceride (MG), phosphatidylcholine (PC), phosphatidylethanolamine (PE), phosphatidylglycerol (PG), triglyceride (TG).

**Figure 2 marinedrugs-22-00067-f002:**
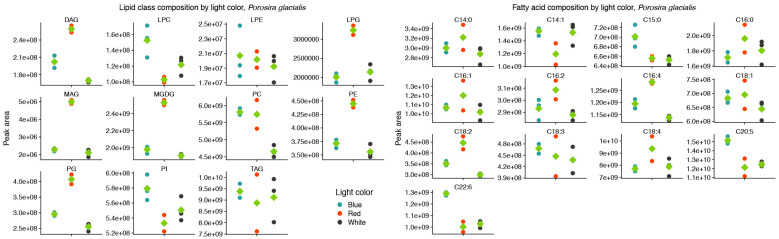
Red light increases the contributions of the thylakoid-associated lipids MGDG and PG and blue light increases the contributions of the long-chain polyunsaturated fatty acids C20:5 and C22:6 in the lipid class (**left**) and fatty acid (**right**) composition of *Porosira glacialis*. Only fatty acids that on average contributed more than 0.1% of the total peak area are included. Diglyceride (DG), lysophosphatidylcholine (LPC), lysophosphatidylethanolamine (LPE), lysophosphatidylglycerol (LPG), monogalactosyldiacylglycerol (MGDG), monoglyceride (MG), phosphatidylcholine (PC), phosphatidylethanolamine (PE), phosphatidylglycerol (PG), phosphatidylinositol (PI), triglyceride (TG). Green diamonds represent mean values, *n* = 3.

**Figure 3 marinedrugs-22-00067-f003:**
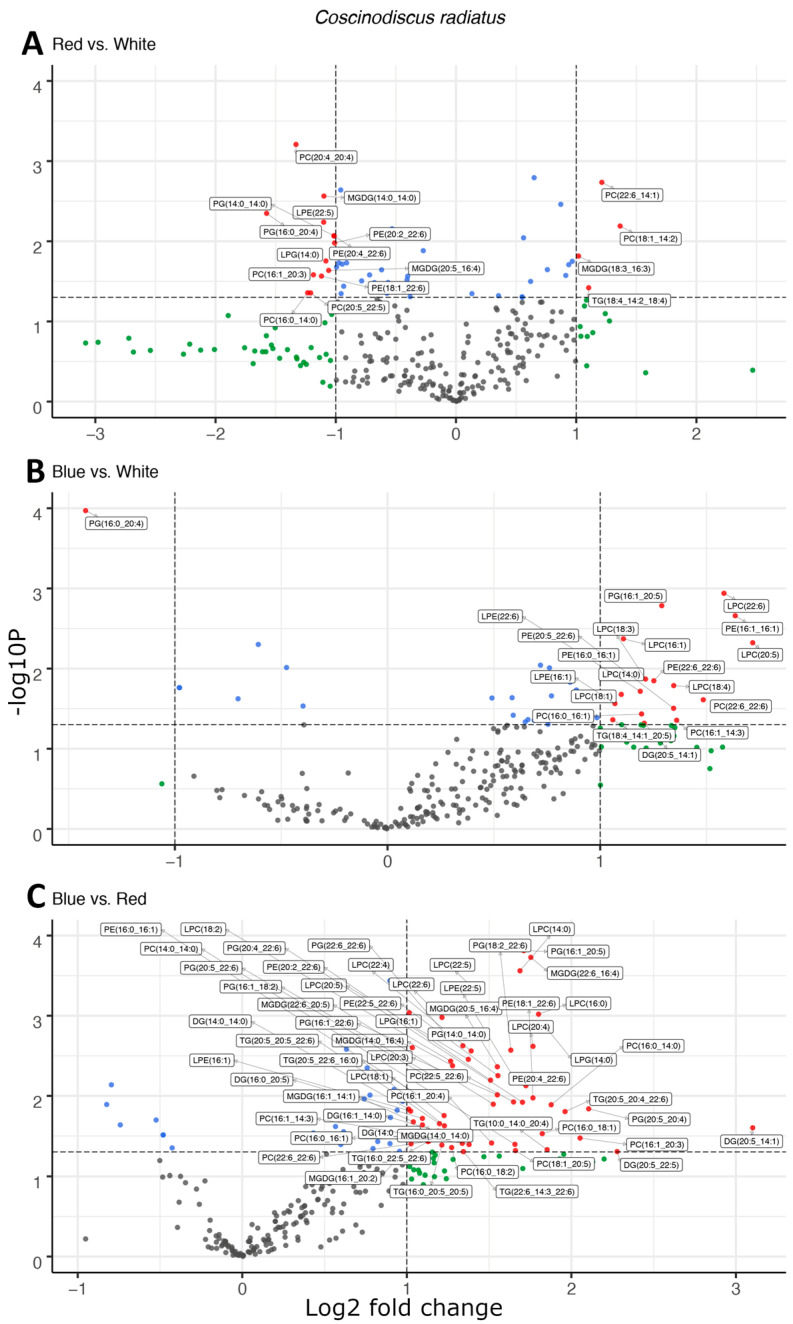
Light quality drives increases the esterification of long-chain polyunsaturated fatty acids in polar, structural lipids observed as the differential abundance of molecular species in volcano plots of red and white light (**A**), blue and white light (**B**), and blue and red light (**C**) in *Coscinodiscus radiatus*. The *y*-axis is defined as the negative log10 of the *p*-value, such that a higher value represents a lower *p*-value. The horizontal and vertical lines represent thresholds to determine significant values (*p* < 0.05 and fold change ≥ 2). The green dots represent data points with a significant fold change, blue dots represent data points with a significant *p*-value and red data points exceed both significance thresholds. Data points represent average peak area for each lipid molecule, *n* = 3. Diglyceride (DG), lysophosphatidylcholine (LPC), lysophosphatidylethanolamine (LPE), lysophosphatidylglycerol (LPG), monogalactosyldiacylglycerol (MGDG), phosphatidylcholine (PC), phosphatidylethanolamine (PE), phosphatidylglycerol (PG), triglyceride (TG).

**Figure 4 marinedrugs-22-00067-f004:**
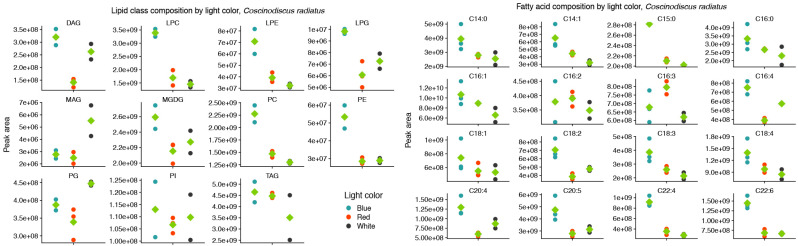
Blue light drives increased inclusion of polar, structural lipids and the long-chain polyunsaturated fatty acids C20:5 and C22:6 in the lipid class (**left**) and fatty acid (**right**) composition of *Coscinodiscus radiatus*. Only fatty acids that on average contributed more than 0.1% of the total peak area are included. Diglyceride (DG), lysophosphatidylcholine (LPC), lysophosphatidylethanolamine (LPE), lysophosphatidylglycerol (LPG), monogalactosyldiacylglycerol (MGDG), monoglyceride (MG), phosphatidylcholine (PC), phosphatidylethanolamine (PE), phosphatidylglycerol (PG), phosphatidylinositol (PI), triglyceride (TG). Green diamonds represent means of peak areas, *n* = 3.

**Figure 5 marinedrugs-22-00067-f005:**
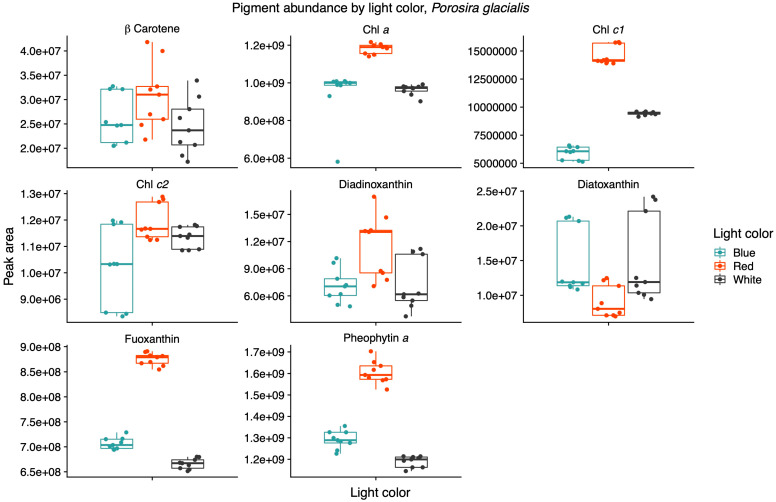
Red light increases the contribution of the main light-harvesting pigments chlorophyll a (Chl *a*), chlorophyll *c1* (Chl *c1*) and fucoxanthin in the psychrophilic diatom *Porosira glacialis*. Mass spectra were imported with both precursor ions and characteristic fragments of the identified pigments using mzVault v2.3 (Thermo Fisher). The most abundant adduct for each pigment standard was considered and the most intense fragmentation spectrum was automatically selected by the software’s algorithmic box plots with overlaid data points. Horizontal lines represent median values, hinges correspond to 25th and 75th percentiles, and whiskers extend to the farthest data point no further than 1.5 × IQR from the hinges, *n* = 9.

**Figure 6 marinedrugs-22-00067-f006:**
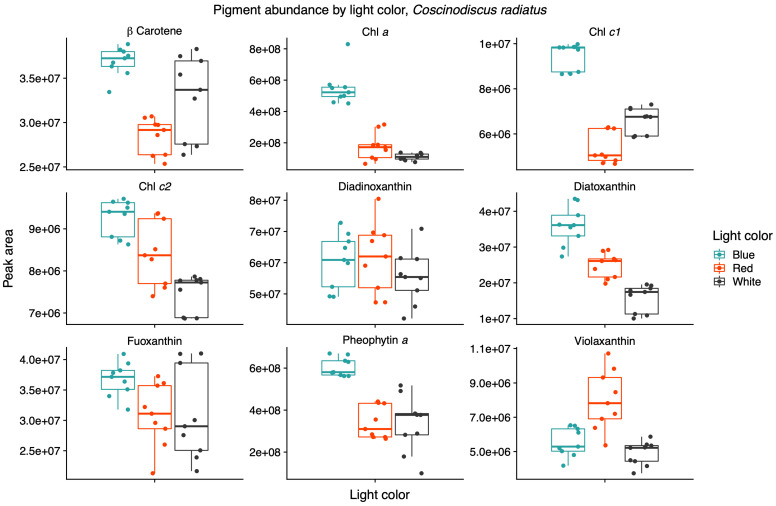
Blue light increases the contributions of chlorophyll a (Chl *a*), chlorophyll *c1* (Chl *c1*), chlorophyll *c2* (Chl *c2*) and diatoxanthin in the mesophile diatom *Coscinodiscus radiatus*. Mass spectra were imported with both precursor ions and characteristic fragments of the identified pigments using mzVault v2.3 (Thermo Fisher). The most abundant adduct for each pigment standard was considered and the most intense fragmentation spectrum was automatically selected by the software’s algorithm. Box plots are overlaid with data points. Horizontal lines represent median values, hinges correspond to 25th and 75th percentiles, and whiskers extend to the farthest data point no further than 1.5 × IQR from the hinges, *n* = 9.

**Figure 7 marinedrugs-22-00067-f007:**
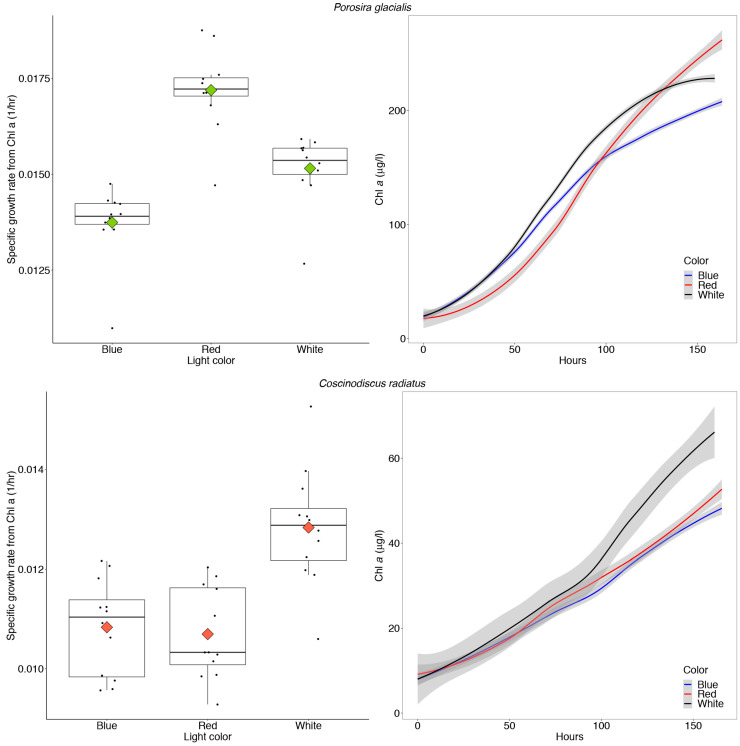
Maximum specific growth rates calculated from in vitro chlorophyll a (Chl *a*) and absolute amounts of in vitro Chl *a* over time in *Porosira glacialis* (**top**) and *Coscinodiscus radiatus* (**bottom**), when cultivated for 7 days at 35 μmol m^−2^ s^−1^ of red, white and blue light, respectively. The box plots include all data points as well as the means (diamonds) and medians (vertical black lines). The error-bars in the box plots and gray shaded error bands in the curve plots represent the SD for each sample, *n* = 12.

**Figure 8 marinedrugs-22-00067-f008:**
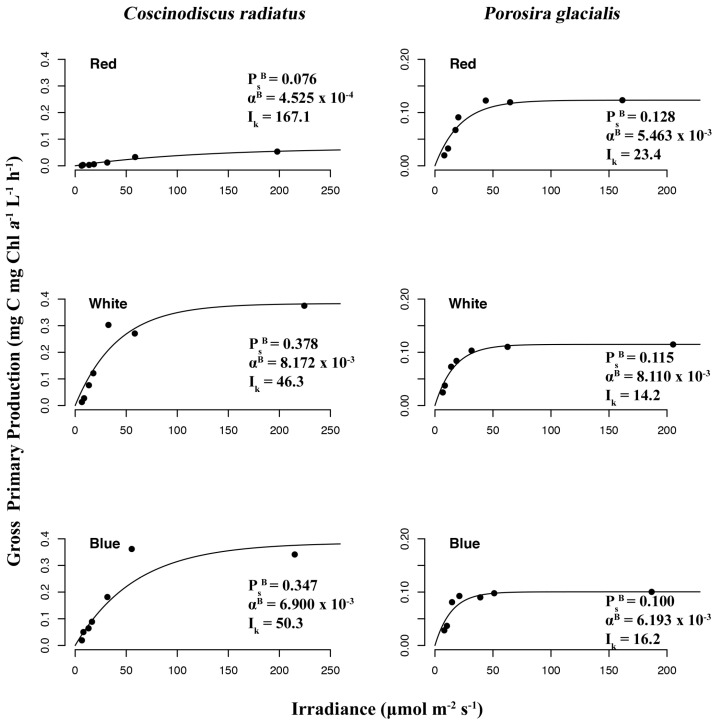
Photosynthesis–irradiance (P–I) curves reveal irradiance wavelength dependency of gross primary production to photon density determined by ^14^C radioactive tracer accumulation in *Porosira glacialis* (**right**) and *Coscinodiscus radiatus* (**left**) in red, blue and white light.

**Table 1 marinedrugs-22-00067-t001:** The total number of data points exceeding both differential abundance significance thresholds (*p* < 0.05, fold change ≥ 2) in volcano plots.

	*P. glacialis*	*C. radiatus*
Red vs. white	22	17
Blue vs. white	4	20
Blue vs. red	37	59

**Table 2 marinedrugs-22-00067-t002:** The average amount of chlorophyll *a* (Chl *a*) per cell at the start of the experiment, following a 7-day acclimatization period to each light quality in *Porosira glacialis* and *Coscinodiscus radiatus*, *n* = 6.

Light Quality	Chl *a*/Cell, *P. glacialis*	Chl *a*/Cell, *C. radiatus*
Red	1.6 × 10^−5^ ± 1.3 × 10^−6^	1.8 × 10^−5^ ± 6.7 × 10^−7^
Blue	1.9 × 10^−5^ ± 1.7 × 10^−6^	1.7 × 10^−5^ ± 4.7 × 10^−6^
White	1.9 × 10^−5^ ± 2.4 × 10^−6^	1.7 × 10^−5^ ± 2.0 × 10^−6^

## Data Availability

The raw data obtained in this study along with the R scripts used for analysis and plotting are available from the Open Science Framework (OSF) under the name “Lipidome plasticity enables unusual photosynthetic flexibility in Arctic vs. temperate diatoms” at https://osf.io/vkhaq/.

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
