# Peer review of "Lipidome Plasticity Enables Unusual Photosynthetic Flexibility in Arctic vs. Temperate Diatoms"

_marinedrugs, 2024, doi:10.3390/md22020067_

Round 1

Reviewer 1 Report

Comments and Suggestions for Authors

The manuscript compared the lipidomic profiles, pigmentation, specific growth rates and carbon assimilation between the Arctic diatom Porosira glacialis versus the temperate species Coscinodiscus radiatus growth under different light colours. In general, while the study is inherently intriguing for shedding light on the differences between diatoms inhabiting regions with stark temperature contrasts, there are several aspects that warrant improvement in the manuscript to enhance its overall quality and clarity.

1. In this study, the author compared two diatoms species originating from Arctic and temperate regions, and attributing their disparities primarily to unique photoadaptation strategy related to variations in light conditions across these regions. However, no genetic background or phylogenetic relationship between the selected diatoms was provided. It is crucial to recognize that genetic disparities, growth temperatures, along with a multitude of other factors, can significantly influence the lipidome and physiological characteristics of phytoplankton. Therefore, I question the author's rationale behind choosing these particular diatoms. Why not opt for two diatom species with closely related genetic backgrounds from the Arctic and temperate environments, or select four (or more) diatoms—two each from Arctic and temperate regions? Alternatively, could the study have been strengthened by including additional temperature treatments, such as subjecting C. radiatus to 8°C and P. glacialis to 20°C? This would potentially enhance the robustness and generalizability of the findings.

2. We know that cellular pigment content can vary significantly across species and under different environmental conditions, especially for this study related to light quality. So I am curious that why the author calculate the growth rate not by cell density but rather by using chl a content. Furthermore, I am curious about the decision not to calculate the chl a content to a per-cell basis, which is often a common practice in such analyses.

3. I was wondering if the author could possibly include details on the light quality within the water column for both Arctic and temperate regions in the Introduction section. This information would serve as a valuable foundation, enabling readers to better understand the lipidome plasticity and other physiological characteristics of the diatoms discussed throughout the manuscript. Furthermore, it would strengthen the ecological significance attributed to these organisms.

4. Line 392. The “-2” and “-1” in the unit of light intensity must be formatted as superscript.

5. Line 400. Why use different starting cell concentration for the two species in this study?

6. Line 459. “Fig. s5” should be ”Fig. S5”.

 Figures and Tables

I suggest the author to rearrange the figures and tables as follows.

7.To enhance readability and facilitate comparison of the two diatoms, Figures 1 and 3 could be merged into a single figure. Concurrently, Table 1 can be transformed into a bar plot and integrated within this new Figure 1. Moreover, formatting issues such as inconsistent font sizes, misaligned axis labels, and the presentation of "-log10(P)" and "log2(fold change)" values should be addressed and rectified.

8. Fig 2. and Fig. 4: There are too many small figures, complex vertical axis text, wide horizontal arrangement of lipid class and fatty class. I suggest that the author could plot these figures in a figure with a log-scaled vertical axis for clarity and consistency. Further, it might be beneficial for the author to attempt combining Figure 2 and Figure 4 into one comprehensive figure. Similar problems for Fig. 5 and Fig. 6.

9. Fig. 7. Please add the description of the grey shadows in the figure legend.

10.Fig. 8. The author could rearrange this figure to remove the empty space.

11.Fig. 8. We know that biological replication is a critical component in scientific experiments, but why the there is no biological replications for this experiment. Moreover, I suggest that the author give details about the chlorophyll content used for the experiment.

12. The statistical analysis was overlooked in figures and is imperative to conduct for this study.

Comments on the Quality of English Language

Minor editing of English language required

Reviewer 2 Report

Comments and Suggestions for Authors

The authors present an interesting work studying the modulation of the liposome, pigments and photosynthesis by the quality of the light in diatoms from temperate regions vs diatoms from the artic, highlighting important modulation, especially relevant for the artic Porosira glacialis. The work as merit, the theme is interesting, and although the framing within the scope of the Journal may not be obvious, I can see it working using “drugs” in a broader sense encompassing bioactive lipids and pigments. This should appeal for researchers working in the fields of Marine Sciences, Microbiology and even others more interested in Climate Change. The methodology is adequate and uses state-of-the-art approaches, namely Omics. Bibliography seems appropriate and up-to-date, and this paper could represent a valuable resource to researchers conducting science in these fields or pursuing the theme further. There are some small issues with the paper that should be addressed before publication, and changes that could make the paper better, but nothing substantial that should prevent its publication in this instance, from my point of view. English in the manuscript is generally good.

 Main comments:

 My main concern regarding this paper is the lack of information about the lipid species actually detected and used for data/statistical analysis. In my opinion, it would be important to include a list (table) of all the lipid species detected/considered in the study (by diatom). If possible, it would be even better to include information about how the lipids were identified/assigned (name of the lipid; calculated mass; shift from calculated mass; RT; MS2/MS3 ions detected confirming identification). This information will be very valuable for other researchers studying the lipidomes of other marine organisms. It would help satisfy my curiosity about the "total of 44 and 45 unique fatty acids in P. glacialis and C. radiatus, respectively". That is a lot of different fatty acids and I would really like to know what they are...

Minor comments:

Line 42: “The structural lipochemistry of diatoms is characterized by polar glycerolipids, and 42 the composition diverges significantly between the non-photosynthetic (or extra-chloro-43 plastic) membranes and the photosynthetic (thylakoid) membranes.” Please add a reference for this statement.

 Line 58: Please define "PAR".

 Line 92: “These two centric diatoms were each cultivated un-92 der red (621 nm), blue (457 nm) and white light, at their respective temperatures typical 93 of Arctic (8 °C) and temperate (20 °C) oceans.” Please add a reference supporting the choice for these temperatures, or base the choices in actual data.

 Line 151: “Lys phosphatidylethanolamine (LPE)”. Replace with “lysophosphatidylethanolamine”.

 Line 153: “Only fatty acids that on average contributed more than 0,1% of the total peak area are included.” Please remove, already include in the text a few lines before.

 Line 161: “The y-axis is defined as the negative log10 of the p-value, so that a higher value represents a lower p-value. The hortizontal and vertical lines represent thresholds to determine significant values (p<0.05 and fold change ≥ 2).” This explanation is not in Figure 1. Include it in both or just in the first figure?

 Line 200: “Light quality influenced the relative pigment content in the two diatoms; red light increased Chl a, Chl c1 and fucoxanthin in the Arctic P. glacialis, and blue light increased Chl a, Chl c1, Chl c2, beta-carotene and diatoxanthin in the mesophilic C. radiatus.” Include reference to figure (5 and 6, respectively). Or simply exclude this sentence, since these results are repeated afterwards, anyway.

 Line 240, Figure 7: “Maximum specific growth rates calculated from in vitro Chlorophyll a (Chl a) and absolute 240 amount of in vitro Chl a over time in Coscinodiscus radiatus (top) and Porosira glacialis (bottom)…” I would change this order, since, up until now, results from Porosira glacialis were always presented before those of Coscinodiscus radiatus, and even in the text that order continues to be respected.

 Line 243: “The error-243 bars in the box plots and grey shaded error bands in the represent the SD for each sample, n=12.” This phrase does not seem to make sense. Please rewrite and clarify.

 Line 252: “…and transitioned towards stationary growth after 160 h…” This is not apparent in the figure. Please clarify.

 Line 276, Figure 8: I would also change the order of this figure with Porosira glacialis graphs on the left, to keep the consistency with other figures, the order in the text and even the order in the caption.

 Line 315: “PC is specifically allocated to the embedded photosystems in higher plants and cyanobacteria…” Do you mean "PG" instead?

 Line 419: “samples were added 20 volumes of DCM/MeOH (2:1 v/v)”. I do not understand what "20 volumes" means. You should rather use volume units (mL) here.

 Line 421: “The lipids were resuspended 421 in 2-propanol to a concentration of 0.5 mg/ml and transferred…” How did you determine lipid quantity to be able to prepare this concentration in 2-propanol? Was it gravimetrically?

 Line 424, 4.5. LC-HRMS analysis: Was the same LC method used for both lipidome and pigment analysis? Were they (both lipids and pigments) analysed on the same LC-MS runs?

 Line 259: “During the method development, a mixed lipid standard (Light SPLASH Lipidomix, Avanti Polar Lipids, Alabaster, AL, USA) containing 13 lipids (15:0-18:1 PC, 18:1 LPC, 15:0-18:1 PE, 18:1 LPE, 15:0-18:1 PG, 15:0-18:1 PI, 15:0-18:1 PS, 15:0-18:1-15:0 TG, 15:0-18:1 DG, 18:1 MG, 18:1 Chol ester, d18:1-18:1 SM, C15 ceramide (d18:1%15:0)) and a mix of MGDG, mainly 16:3-18:3 and 18:3-18:3 (Avanti Polar Lipids) was used to confirm identification in LipidSearch.” This is not very clear... Was the mixture of standards added to every samples? Was any kind of normalization made using the standards, namely when calculating the total amounts per lipid class?

 Line 462: Is "Chol" the abbreviation of cholesterol? Please define.

 Line 462: Is “SM” an abbreviation for sphingomyelin? Please define.

 Line 624, References: Please check the references for consistent presentation. The names of species in the titles are sometimes in italic, other not, and sometimes the year of publication is not in bold (40, 44, 45, 46).

Round 2

Reviewer 1 Report

Comments and Suggestions for Authors

My questions have been well addressed. I recommand this manuscript to be accepted for publication.